# Cassava Foliage Effects on Antioxidant Capacity, Growth, Immunity, and Ruminal Microbial Metabolism in Hainan Black Goats

**DOI:** 10.3390/microorganisms11092320

**Published:** 2023-09-15

**Authors:** Mao Li, Xuejuan Zi, Renlong Lv, Lidong Zhang, Wenjun Ou, Songbi Chen, Guanyu Hou, Hanlin Zhou

**Affiliations:** 1Key Laboratory of Ministry of Agriculture and Rural Affairs for Germplasm Resources Conservation and Utilization of Cassava, Key Laboratory of Ministry of Agriculture and Rural Affairs for Crop Gene Resources and Germplasm Enhancement in Southern China, Tropical Crops Genetic Resources Institute, Chinese Academy of Tropical Agricultural Sciences, Danzhou 571737, China; limaohn@163.com (M.L.); lvrenlong@aliyun.com (R.L.); wenjunou@catas.cn (W.O.); songbichen@catas.cn (S.C.); 2Zhanjiang Experimental Station, Chinese Academy of Tropical Agricultural Sciences, Zhanjiang 524000, China; 3Key Laboratory of Ministry of Education for Genetics and Germplasm Innovation of Tropical Special Trees and Ornamental Plants, Key Laboratory of Germplasm Resources of Tropical Special Ornamental Plants of Hainan Province, School of Tropical Agriculture and Forestry, Hainan University, Danzhou 571737, China; zixuejuan@163.com (X.Z.); lidongzhang@catas.cn (L.Z.)

**Keywords:** cassava foliage, antioxidant, immunity, ruminal bacterial, metabolites

## Abstract

Cassava (*Manihot esculenta* Crantz) foliage is a byproduct of cassava production characterized by high biomass and nutrient content. In this study, we investigated the effects of cassava foliage on antioxidant capacity, growth performance, and immunity status in goats, as well as rumen fermentation and microbial metabolism. Twenty-five Hainan black goats were randomly divided into five groups (*n* = 5 per group) and accepted five treatments: 0% (T1), 25% (T2), 50% (T3), 75% (T4), and 100% (T5) of the cassava foliage silage replaced king grass, respectively. The feeding experiment lasted for 70 d (including 10 d adaptation period and 60 d treatment period). Feeding a diet containing 50% cassava foliage resulted in beneficial effects for goat growth and health, as reflected by the higher average daily feed intake (ADFI), average daily gain (ADG) and better feed conversion rate (FCR), as well as by the reduced serum levels of alanine aminotransferase (ALT), aspartate aminotransferase (AST), creatinine (CRE), and triglycerides (TG). Meanwhile, cassava foliage improved antioxidant activity by increasing the level of glutathion peroxidase (GSH-Px), superoxide dismutase (SOD), and total antioxidant capacity (T-AOC) and lowering malondialdehyde (MDA). Moreover, feeding cassava foliage was also beneficial to immunity status by enhancing complement 3 (C3), complement 4 (C4), immunoglobulin A (IgA), immunoglobulin G (IgG), and immunoglobulin M (IgM). Furthermore, the addition of dietary cassava foliage also altered rumen fermentation, rumen bacterial community composition, and metabolism. The abundance of *Butyrivibrio*_2 and *Prevotella*_1 was elevated, as were the concentrations of beneficial metabolites such as butyric acid; there was a concomitant decline in metabolites that hindered nutrient metabolism and harmed host health. In summary, goats fed a diet containing 50% cassava foliage silage demonstrated a greater abundance of *Butyrivibrio*_2, which enhanced the production of butyric acid; these changes led to greater antioxidant capacity, growth performance, and immunity in the goats.

## 1. Introduction

It is well established that forage is an important dietary resource for ruminants that is critical for ensuring rumen health and improving production. In recent years, ruminant husbandry has rapidly developed in China, and the total amount of ruminant livestock raised has greatly increased; the existing grassland area was limited, leading to a shortage of high-quality forage [1,2,3]. As a result, large quantities of forage are imported every year, increasing feed costs and affecting the sustainability of ruminant husbandry; the importation of feed has become a key constraint on the development of animal husbandry in China [1]. Goats are important ruminant livestock in the tropical and subtropical regions of China, but cash crops were the dominant plant in these regions; the lack of native grass brought about insufficient roughage supplies [3]. The pursuit of better exploitation of locally available forage resources has caused an upsurge in related research.

Cassava (*Manihot esculenta* Crantz) is an important crop in sub-tropics and tropics worldwide [4,5]. Cassava foliage is a byproduct of cassava production and is characterized by high biomass, digestibility and protein content, as well as low fiber content [6,7,8]. Therefore, cassava foliage represents an ideal feed source with the potential to aid the sustainable development of local animal husbandry in the tropical regions of the world. In less developed countries and regions lacking high-quality feed, adding cassava foliage to animal diets has a positive effect on digestion, growth, and gastrointestinal tract development [9,10,11,12,13,14]. However, for each ruminant species, there is an optimum percentage of cassava foliage in the diet to maximize growth; for example, the ideal is 75% for sheep, 60% for West African Dwarf goats, 50% for pigs, and 5% for geese [9,10,13,15]. Based on the above studies, cassava foliage should support goat production in tropical China, but the ideal ratio of cassava foliage supplementation remains unknown.

Previous studies have primarily focused on production without examining effects on the gut microbiota and metabolite diversity. Microbial communities in ruminant digestive tracts play a key role in forage digestion and absorption, converting plant material into metabolites such as amino acids, ammonia, peptides, and short-chain fatty acids (SCFAs) [16,17]. Many studies have shown that diet composition regulates ruminant growth and development by affecting the rumen microbiota and associated metabolites [18,19,20]. In previous studies [13,21], the addition of cassava foliage to the diet of geese affected intestinal microbial diversity and gene function, promoting growth performance. However, the effects of cassava foliage addition on rumen microecology remain unknown. To more efficiently utilize cassava foliage in animal diets, understanding its effects on gut microorganismal community composition and metabolism will be essential for determining precision animal nutrition. Thus, this study investigated the effects of cassava foliage addition on antioxidant capacity, growth performance, immunity, and ruminal microbial metabolism in goats from tropical China.

## 2. Materials and Methods

The animal experiments in this study were approved by the Animal Care and Use Committee of the Chinese Academy of Tropical Agricultural Sciences (No. CATAS-20140101). All experimental procedures were performed in agreement with relevant guidelines.

### 2.1. Animal, Diet, and Sample

Twenty-five Hainan black goats of similar body condition (10.20 ± 0.86 kg) were randomly assigned to five treatment groups (*n* = 5 per group). Animals were housed individually and offered free access to fresh water. Their diet consisted of 50% goat feed and 50% forage; feed was provided twice daily at 07:30 and 17:00. The dietary ingredients and nutrition composition were formulated to meet NRC (2007) recommendations and previously described nutritional requirements [22]. The same amount of goat feed and forage was provided to each treatment group, but the composition of the forage varied by treatment. The forage contained varying amounts of cassava foliage silage in place of king grass: 0% (T1), 25% (T2), 50% (T3), 75% (T4), and 100% (T5). The chemical composition of the goat diet (including concentrate, cassava foliage silage, and king grass) for each treatment group is described in Table 1. The feeding experiment lasted for 70 days (including a 10 day adaptation period and a 60 day treatment period).

The measurement of average daily gain (ADG), average daily feed intake (ADFI), and the feed conversion ratio (FCR) were conducted as previously described; the details are as follows: each goat was weighed on the first and last day of the treatment period to calculate ADG, amounts of feed provided and the residual of each goat were recorded daily during the treatment period to calculate DMI, and the FCR was the ratio of ADFI:ADG [3]. About 4 mL of blood samples were taken from each goat’s jugular vein by applying a vacutainer (no additive) before morning feeding at 60 days of the treatment period. The above samples were centrifuged at 3000× *g* for 20 min at 4 °C, and the serum was separated into three parts and stored at −20 °C for biochemical indexes, antioxidant capacity, and immunity factors analyses. The protocol was according to methods described by Li et al. [3] and Wang et al. [19]. Ruminal fluid samples were collected one hour before the morning feeding on the last day of the animal experiment. The pH of the fluid samples was measured immediately after collection using a pH meter. The fluid samples were then divided and stored at either −20 °C for later assessment of the rumen fermentation index or −80 °C for later analysis of ruminal bacterial communities and their metabolites [3,19].

### 2.2. Chemical Analysis

Rumen fermentation characteristics were measured via gas chromatography (GC-2014B, Shimadzu, Kyoto, Japan), while blood-based biochemical indicators were analyzed using commercially available kits (Sigma Aldrich, St. Louis, MO, USA) according to the manufacturer’s instructions with an automatic biochemical instrument (PUZS-600B, Langpu New Technology, Co., Ltd., Beijing, China). Both antioxidant and immune indicators were evaluated using commercial kits (Nanjing Jiancheng Biotech, Nanjing, China). The above assessments followed protocols described by Li et al. [23] and Wang et al. [19].

### 2.3. Microbial Diversity Analysis

The bacterial communities within ruminal fluid samples were characterized by extracting microbial DNA using the E.Z.N.A.^®^ soil DNA Kit (Omega Bio-Tek, Norcross, GA, USA). The DNA quantity and quality were detected by a NanoDrop 2000 UV-Vis spectrophotometer (Thermo Scientific, Wilmington, NC, USA). The 338F and 806R primers were used to amplify the V3–V4 region of DNA on a standard PCR thermocycler (GeneAmp 9700, ABI, Waltham, MA, USA). The high-quality PCR products were then sequenced on an Illumina MiSeq 2500 platform (Illumina, Inc., San Diego, CA, USA) with paired-end 250 bp reads [2].

Raw sequence data were processed using MOTHUR [24]. Clean reads were clustered into operational taxonomic units (OTUs) using a confidence threshold of 97%. Alpha diversity (i.e., OTU number and Shannon/Simpson’s diversity indices) and beta diversity (as determined via non-metric multidimensional scaling [NMDS]) were quantified using QIIME2. Rumen bacterial community structure was analyzed at the phylum and genus levels using the Silva database (version 138), with a similarity cut-off of 70%. To identify rumen bacteria unique to each dietary group, the linear discriminant analysis effect size (LefSe) tool was used with a linear discriminant analysis (LDA) score greater than 4.0 [25]. Bioinformatics analyses were performed in BMKCloud (Biomarker Technologies Co., Ltd., Beijing, China). All raw sequence data were deposited into the NCBI Sequence Read Archive (SRA) database (No. PRJNA951504).

### 2.4. Metabolome Analysis

The ruminal fluid samples were combined with pre-chilled methanol and L-2-chlorophenylalanine in tubes. These were mixed well on a vortex, and mixtures were then ultrasonicated and centrifuged. The supernatant was collected, and a QC (quality control) sample was prepared; the remaining samples were then pooled and evaporated in a vacuum concentrator. Next, the pooled samples were incubated with methoxyamine hydrochloride and derivatized with BSTFA. The QC sample was cooled to room temperature, and then a mixture of fatty acid methyl esters (FAMEs) was added to identify metabolites. Gas chromatography paired with time-of-flight mass spectrometry (GC-TOF-MS) was performed using an Agilent 7890 gas chromatographer coupled with a time-of-flight mass spectrometer. The analyzed system used a DB-5MS capillary column. Instrument parameters used here follow previously reported protocols [18].

Raw data were analyzed in Chroma TOF (V 4.3x, LECO, St. Joseph, MI, USA), and metabolite identification was performed using the LECO-Fiehn Rtx5 database. Peaks with a relative standard deviation (RSD) of >30% or <50% in the QC samples were removed [26]. In a follow-up metabolite data analysis, all samples were analyzed using principal components analysis (PCA) and partial least-squares discriminant analysis (PLS-DA). Identified compounds were classified, and pathway information was obtained from the KEGG database. Differentially expressed metabolites (DEMs) (*p* < 0.05) were identified using Student’s *t*-tests and an orthogonal partial least-squares discriminant analysis (OPLS-DA) based on variable importance in projection (VIP) values greater than one. Differentially expressed metabolites in the KEGG pathway enrichment analysis were identified using hypergeometric distribution tests. Heatmaps were generated using the ‘corrplot’ package in R and BMKCloud (www.biocloud.net, accessed on 3 April 2023).

The above procedures for sample preparation, metabolite separation and identification, and data analysis are described in greater detail in a previous publication [18].

### 2.5. Statistical Analysis

After the Shapiro–Wilk test, all data in this study’s distributions accord with normal distribution; the impact of cassava foliage inclusion on growth performance, blood biochemical, antioxidant, immune, and rumen fermentation indexes of goats were investigated using one-way analyses of variance as implemented in SAS v. 9.3 (SAS Institute Inc., Cary, NC, USA). The orthogonal polynomial contrasts (linear and quadratic) were used to evaluate the effect of the cassava foliage inclusion ratio in the goat diet. Duncan’s multiple range tests were used to identify significant differences, and effects with *p* < 0.05 were considered statistically significant.

## 3. Results

### 3.1. Growth Performance

The effects of the diet treatments on goat growth performance are shown in Table 2. The ADG was significantly higher for goats on a diet including cassava foliage than for those without (*p* < 0.05), but ADG did not differ among goats from T2, T3, T4, and T5 (*p* > 0.05). The ADFI was highest in the T3 treatments group (*p* < 0.05) and showed a trend of first increasing and then decreasing. The FCR was significantly lower for goats consuming cassava foliage than for controls (i.e., no cassava foliage included in diet) (*p* < 0.05), and the four treatment groups (i.e., all diets including cassava foliage) had similar FCR, which meant that added cassava foliage improves feed conversion efficiency.

### 3.2. Blood Biochemical Indexes

As shown in Table 3, the addition of cassava foliage to the diet decreased the concentration of alanine aminotransferase (ALT), aspartate aminotransferase (AST), creatinine (CRE), and triglyceride (TG) (*p* < 0.05); the concentrations were highest in T1 and lowest in T5 (*p* < 0.05). However, the concentration of albumin (ALB), blood urea nitrogen (BUN), total cholesterol (TCHO), total protein (TP), and uric acid (UA) did not differ among treatments (*p* > 0.05).

### 3.3. Oxidative Status

Four antioxidant indexes (indicators of overall oxidative status) are shown in Table 4. The concentration of GSH-Px, T-AOC, and T-SOD first increased and then decreased with the addition of greater proportions of cassava foliage to the diet (*p* < 0.05). In contrast, the MDA concentration showed the opposite pattern. Comparing treatments, T3 had the highest oxidative capacity.

### 3.4. Immunity Status

Table 5 shows the immunity status of the study goats. Concentrations of C3, C4, IgA, IgG, and IgM first increased and then decreased with the proportion of cassava foliage added to the diet (*p* < 0.05). The moderate treatment (T3; 50% cassava foliage) had the highest immune capacity.

### 3.5. Rumen Fermentation Index

Rumen fermentation index values for goats in different treatments are provided in Table 6. Rumen pH and levels of acetate, isobutyrate, isovalerate propionate, and valerate were similar among treatments (*p* > 0.05). The ruminal butyrate concentration was highest in T2 (*p* < 0.05) and lowest in T5 (*p* < 0.05). The butyrate concentration decreased as the proportion of cassava foliage increased (i.e., from T2 to T3, T4 and T5).

### 3.6. Diversity of the Rumen Microbiome

The α-diversity analysis results of the rumen bacterial communities are shown in Figure 1, including cassava foliage in the diet affected the Shannon and Simpson diversity indices for rumen bacterial communities (Figure 1A,B). The Shannon diversity index was reduced in T3, and the Simpson diversity index was higher in T4 vs. T3 (*p* < 0.05); this suggests that consuming cassava foliage resulted in lower α-diversity. A total of 915 OTUs were identified in the rumen microbiome. The five treatment groups had 807 OTUs in common, and only one OTU was unique to the T5 group (Figure 1C). In the NMDS analysis, the rumen microbial communities varied significantly among the five treatments (Figure 1D), suggesting that microbial community structure shifted in response to cassava foliage consumption.

The composition of the rumen bacterial communities was also compared among treatments (Figure 2). Bacteroidetes and Firmicutes were the two predominant phyla identified. Bacteroidetes abundance increased with the proportion of cassava foliage, while Firmicutes abundance decreased (Figure 2A). The *Prevotella*_1 and *Rikenellaceae*_RC9_gut_group were the most common genera in the rumen bacterial communities (Figure 2B). The abundance of both *Butyrivibrio*_2 and *Prevotella*_1 significantly increased with the proportion of cassava foliage (*p* < 0.05). In addition, the abundance of unclassified bacteria decreased from 25.38% (T1) to 20.53% (T5).

Differences in the rumen bacterial communities among treatments were detected using the LEfSe method, and the microbial taxa specific to each group were identified (Figure 2 C,D). Some biomarkers were enriched in the five groups. The genus *Fretibacterium*, belonging to the family *Synergistaceae*, order Synergistales, class Synergistia, and phylum Synergistetes, and genus *Quinella*, belonging to the family *Veillonellaceae*, were the microbial taxa enriched in T1. *Lactobacillus acetotolerans*, which belongs to the order Lactobacillales and class Bacilli, and the family *Lachnospiraceae*, which belongs to the phylum Firmicutes, were enriched in T2. The genus *Prevotellaceae*_UCG_003 and family *Prevotellaceae* were enriched in T3. The genus *Candidatus Saccharimonas* was enriched in T4. In addition, the microbial taxa enriched in T5 included the order Bacteroidales, class Bacteroidia, and phyla Bacteroidetes.

### 3.7. Analysis of Rumen Metabolites

The metabolomic analysis yielded 4913 features. These were used as a batch query against the human metabolome database (HMDB) to annotate 4330 individual samples with identified features. The identified metabolites were further divided into eight superclasses and 20 categories (Figure 3) as follows: benzenoids (benzene and its substituted derivatives and phenol), eterocyclic compounds (quinolines and derivatives lipids and lipid-like molecules), lipids and lipid-like molecules (fatty acyls, glycerolipids, glycerophospholipids, prenol lipids, sphingolipids, steroids, and steroid derivatives), organic acids and derivatives (carboxylic acid and derivatives), organic nitrogen compounds (organonitrogen compounds), organic oxygen compounds (organooxygen compounds), organoheterocyclic compounds (benzopyran, indoles and derivatives, pteridines and derivatives, pyridines and derivatives, and quinolines and derivatives), phenylpropanoids and polyketides (cinnamic acid and derivatives, coumarin and derivatives, and flavonoids).

A comparative metabolomic analysis was used to determine how rumen metabolites differed between goats fed diets with or without cassava foliage, finding significant differences in their metabolic profiles (Figure 4). There were 881, 1782, 1189, and 1825 differentially expressed metabolites (418, 791, 651, and 1035 up-regulated and 463, 991, 538, and 790 down-regulated metabolites) obtained in the goats fed diets without cassava foliage vs. those fed diets with cassava foliage (for T1 vs. T2, T1 vs. T3, T1 vs. T4, and T1 vs. T5, respectively) (Figure 4). The ten differentially expressed metabolites with the highest fold changes could be used as potential biomarkers of cassava-based diets (Figure 5). The most highly up-regulated metabolites included sorbitan stearate, lysoSM (d18:1), (S)-oleuropeic acid, 5-dehydroavenasterol, and naziminin A, while the most significantly downregulated metabolites included glutaric acid, 5-hydroxy-2-furoic acid, norfuraneol, and D-glucuronolactone.

A KEGG pathway enrichment analysis of the differentially expressed metabolites revealed that diets including cassava foliage showed altered microbial community functioning. A number of metabolic pathways were significantly enriched in T1 vs. T2 (*n* = 8), T1 vs. T3 (3), T1 vs. T4 (2), and T1 vs. T5 (3) (Figure 6). The eight significantly enriched metabolic pathways in T1 vs. T2 included flavonoid biosynthesis, amino sugar and nucleotide sugar metabolism, protein digestion and absorption, cysteine and methionine metabolism, glycolysis/gluconeogenesis, prodigiosin biosynthesis, propanoate metabolism, and Type I polyketide structures (Figure 6A). The metabolic pathways of benzoate degradation, neuroactive ligand-receptor interaction, and steroid biosynthesis were significantly enriched in T1 vs. T3 (Figure 6B). Meanwhile, lysine degradation and type I polyketide structure pathways were significantly enriched in T1 vs. T4 (Figure 6C). Finally, polycyclic aromatic hydrocarbon degradation, biosynthesis of siderophore group nonribosomal peptides, and serotonergic synapse pathways were significantly enriched in T1 vs. T5 (Figure 6D).

## 4. Discussion

Cassava foliage is rich in nutrients and highly digestible [6,7]. The ADFI, ADG, and FCR are important indicators of animal feed value. Here, goat diets containing cassava foliage promoted growth, consistent with effects seen for ruminants in West Africa [9,15]. However, it has also been reported that cassava leaves contain antinutrients (e.g., hydrogen cyanide [HCN]), which may affect animal health and growth [27]. We found that ADFI decreased as the proportion of cassava foliage increased (beyond 50%), while ADG and FCR did not vary. This suggests that there was a limit to the benefits of adding cassava foliage to the diet. Similarly, goose performance was higher on diets containing 5% vs. 10% cassava foliage [13]. In conclusion, different animal species vary in their ability to thrive on diets containing cassava foliage, and identification of the optimum proportion of cassava foliage in the diet is necessary.

In general, blood biochemical indexes reflect whether nutrient digestion and metabolism, as well as tissue and organ functioning, are normal; these indexes can, therefore, be used as indicators of animal health and for the diagnosis of abnormalities [1]. Clinical detection of serum CRE is one of the methods commonly used to understand renal function. Reynolds et al. [28] observed similar CRE levels in goats fed with tannin-rich pine bark vs. Bermuda grass. Similarly, Li et al. [13] reported comparable CRE concentrations in geese fed different amounts of cassava foliage. However, in the present study, the CRE concentrations in groups T3, T4, and T5 were lower than those in T1 and T2; this suggests that higher proportions of cassava foliage in the diet (50%+) may damage renal health. TG represents the largest lipids in animal bodies and a primary form of energy storage; their concentration reflects liver lipid metabolism. Reynolds et al. [28] and Li et al. [3] reported that roughage type had no influence on TG content. Nevertheless, here, diets containing cassava foliage significantly reduced TG content. This suggests that cassava foliage may have a role in regulating liver lipid metabolism. ALT and AST are primarily found in liver and heart tissue cells. When these tissues become diseased, local enzyme activity increases, leading to elevated concentrations of ALT and AST. In this study, ALT and AST concentrations declined as greater proportions of cassava foliage were added to the diet. This decrease in ALT and AST may imply that cassava foliage had a positive effect on heart and liver health. However, this result is inconsistent with previous studies (e.g., [3,28]), where ALT and AST concentrations did not vary among goats fed different forages. This discrepancy may be related to the types of roughage used and the tolerance of the animals, and further research is needed. Collectively, considering its influence on the health and functioning of the heart, kidneys, and liver, the proportion of cassava foliage in tropical Chinese goat diets should not exceed 50%.

Numerous studies have reported that cassava foliage contains multiple bioactive compounds, including bioactive flavonols (e.g., apigenin, kaempferol, and rutin) and phenolics [29,30,31]. Therefore, animal diets, including cassava foliage, can improve antioxidant capacity. In a previous study of geese [14], greater cassava foliage content in the diet enhanced antioxidative status. Similarly, cassava foliage intake can increase antioxidant action in chickens [32]. Similar phenomena have been found in mammals. For example, ethanolic extracts of cassava leaves significantly increased antioxidant enzyme serum levels in Wistar rats [33], and piglets fed cassava residues also showed elevated antioxidant capacity [31]. We observed that the antioxidant capacity first increased and then decreased as the proportion of cassava foliage increased in the diet; the 50% cassava foliage group showed the highest oxidative capacity. This discrepancy (as compared to other studies) may be attributable to variations in dietary composition (e.g., in the supplements included and the amount of cassava foliage), as well as in the digestive capacity of the study animals.

Organismal immunity status refers to the ability of natural defense mechanisms to resist disease. To evaluate how cassava foliage impacts immune function in goats, immunoglobulin levels were assessed. Cassava foliage inclusion raised the immunity status of goats by increasing the value of immune indexes. In line with this result, earlier studies have reported that feeding piglets cassava residues increased IgA levels and improved immunomodulatory functions [31]. Moreover, the use of fermented feed may improve animal immune system functioning. Similarly, alfalfa silage promoted higher immunity status in lactating dairy goats [34]. Fermented feed may improve immune ability by supporting beneficial microorganisms and their metabolites. In conclusion, the inclusion of cassava foliage can affect immunity status, but this relationship is dose-dependent and needs to be carefully researched.

Volatile fatty acids (VFAs) in the rumen are produced by the microbial fermentation of carbohydrates and can be used by ruminants to meet energy demands. The main VFA components are acetate, propionate, and butyrate, with acetate accounting for the largest proportion [34]. According to our data, the inclusion of cassava foliage inclusion did not affect rumen pH nor the concentration of acetate or propionate but significantly affected the butyrate concentration. As more cassava foliage was included in the diet, butyrate levels decreased. This is consistent with the results of Harun et al. [35], who studied the effects of cassava foliage on in vivo rumen fermentation; butyrate concentrations were lower in cassava-foliage-fed groups, but other VFAs were unaffected. Butyrate plays an essential role in cellular energy metabolism and the induction of apoptosis, as well as in regulating immune function, inflammatory responses, and intestinal homeostasis [36,37]. Meanwhile, butyrate has also been shown to promote growth in ruminants and to enhance dairy product quality [38,39]. Therefore, understanding the rumen microbial community structure, especially the abundance and composition of butyrate-producing bacteria, will be essential for optimal ruminant production.

The determination of rumen microbial community composition is helpful not only for understanding ruminant physiology but also for the precise management of animal nutrition to improve feed conversion efficiency [16]. Lately, many studies have shown how diet can regulate rumen microbial community composition and metabolism; the rumen microbiome has been linked not only to the host diet but also to host growth performance, immune function, and physiological status, among other phenotypes [19,20,37]. For example, Wang et al. [40] observed that the roughage type had a significant impact on rumen microorganisms and metabolites, thereby altering growth performance. Cassava foliage has been widely used in ruminant diets because of its positive effect on production. However, how cassava foliage affects rumen microbial communities remains poorly understood. In the current study, an integrated approach of 16S rRNA sequencing and GC-MS-based untargeted metabolomics was applied to examine the goat rumen microbiome and metabolome in order to assess any effects of cassava foliage inclusion in the diet. Cassava foliage inclusion treatments resulted in lower bacterial diversity and richness, as well as significant differences in bacterial composition. However, these results are not consistent with studies of intestinal microbes in monogastric animals. For example, the inclusion of cassava foliage (5% by weight) in the diet of geese elevated their intestinal microbial diversity [21]. In addition, supplementation with fermented cassava residues in the diet of piglets did not significantly alter gut microbial diversity [31]. Therefore, ruminants and monogastric animals must have fundamental differences in their digestive capacity.

As in previous studies, *Prevotella* had the highest abundance in goat rumen microbial communities; *Prevotella* is closely linked to the digestion and metabolism of fiber and proteins [20,41,42]. Hence, *Prevotella* represents a core microbial taxa in rumen communities. In the rumen, *Prevotella* degrades and utilizes starch and plant cell wall polysaccharides, such as pectin and xylan, producing large amounts of SCFAs for the host [43,44]. Comparing sheep fed corn silage vs. corn stalks [45], those fed corn silage showed a greater abundance of *Prevotella*, as well as differences in fiber and protein levels. In this study, similar results were observed: *Prevotella* abundance first increased with the proportion of cassava foliage in the diet before decreasing. This could be due to changes in the fiber-to-protein ratio in the diet, leading to adaptive variation in *Prevotella* abundance to more efficiently extract nutrients from roughage.

We also demonstrated that the *Rikenellaceae*_RC9_gut_group (family *Rikenellaceae*) was the second most common bacterial taxa in the goat rumen. *Rikenellaceae* has been linked to the degradation and absorption of structural carbohydrates, producing acetic and propionic acid; it ensures that the rumen maintains an appropriate ratio of short-chain fatty acids [20,46]. Yang et al. [47] and Zhang et al. [20] found that rumen *Rikenellaceae* abundance was positively correlated with the digestion, absorption, and metabolism of nitrogen, leading to downstream effects on animal production. Similarly, Li et al. [34] found that a greater abundance of *Rikenellaceae* supported sheep fattening. However, the abundance of *Rikenellaceae* among groups was very similar, but there were clear differences in growth production. This may be due to divergence among animal species in *Rikenellacae* abundance.

Here, an interesting finding was that the abundance of *Butyrivibrio*_2 was significantly higher in the cassava foliage addition treatments. *Butyrivibrio* exists in human intestines and animal rumens; it ferments carbohydrates and cellulose to produce butyric acid, which plays an essential role in regulating antioxidant capacity, cellular energy metabolism, and immunity, as well as in protecting the structural morphology and functional stability of the intestinal epithelium [37,39]. *Butyrivibrio* utilizes higher cellulose or lignocellulosic in ruminants and humans [39,48]. Liu et al. [49] found that *Buryrivibrio* was more abundant in yaks than cattle, which could explain the higher digestibility of fiber for yaks vs. cattle. However, the presence of other bioactive compounds in the diet might also affect *Buryrivibrio* abundance. Consistent with study results, Wang et al. [19] reported higher rumen butyric acid levels and *Butyrivibrio* abundance in dairy cows fed with inulin, a polysaccharide that is particularly concentrated in Jerusalem artichoke (*Helianthus tuberosus* L.) tubers. In conclusion, understanding how cassava foliage-based diets alter rumen *Buryrivibrio* abundance will require more in-depth research.

Sorbitan stearate is used in the food manufacturing industry as a food emulsifier, stabilizer, and flavor modifier [50]. LysoSM (d18:1) is considered an intermediate in sphingolipid metabolism, which has generally been linked to human heart health [51,52]. Dehydroavenasterol belongs to the class of organic compounds known as stigmastanes and derivatives and is an intermediate in the biosynthesis of steroids, as well as a participant in lipid metabolism [53]. Niaziminin A belongs to the class of organic compounds known as phenolic glycosides and represents one of the main physiologically active components found in *Moringa oleifera* [54,55]. These up-regulated metabolites are related to either the digestion and metabolism of nutrients or the anabolism of bioactive substances, illustrating how feeding cassava foliage could promote nutrient utilization and host health.

Glutaric acid is an end product of organism metabolism of certain amino acids (e.g., lysine and tryptophan); high levels of glutaric acid are associated with metabolic issues, causing adverse health effects [56]. Consistent with the present study, Zhang et al. [20] also found that the dietary protein level affected the glutaric acid concentration of the rumen. Furoic acid is a metabolite and marker of host exposure to furfural, a confirmed carcinogen dangerous to animal health [57]. Norfuraneol has been detected in several different foods, such as beer, blackberries (*Rubus* spp.), evergreen blackberries (*Rubus laciniatus*), and various fruits. Thus, norfuraneol may be a potential biomarker for the consumption of these foods. Glucuronolactone is a key structural component of plant connective tissues that are routinely referred to as a natural substance; it has been shown to ameliorate liver injury [58]. Meanwhile, glucuronolactone is a popular ingredient in energy drinks because it can effectively increase energy levels and improve alertness [59]. These results suggest that feeding cassava foliage could decrease the abundance of metabolites that hinder nutrient metabolism, are hazardous to host health, and limit structural carbohydrate utilization.

Previous studies have also shown that diet can affect rumen metabolite composition and alter metabolic pathways. For example, Li et al. [34] found that the inclusion of pelleted TMR in the diet up-regulated amino acid metabolism and steroid biosynthesis in lambs, contributing to better production. Wang et al. [19] observed differentially enriched metabolic pathways in dairy cows fed dietary inulin supplements; these pathways included amino acid metabolism, vitamin metabolism, nucleotide metabolism, and plant secondary metabolites biosynthesis. These results are generally consistent with the findings of our study but identify different pathways. However, Zhang et al. [20] reported that higher dietary protein levels significantly affected the TCA cycle pathway, which is a key pathway for the synthesis and/or conversion of glucose, amino acids, and fatty acids. In summary, both dietary and functional ingredients had an important influence on the metabolic pathways identified for differentially expressed metabolites.

## 5. Conclusions

Overall, feeding goats a diet containing 50% cassava foliage silage resulted in beneficial effects on animal health and performance, as reflected by higher ADFI, ADG, and feed conversion efficiency. A cassava foliage-supplemented diet also enhanced the antioxidant activity and immunity status of goats. Moreover, the addition of cassava foliage altered rumen fermentation, rumen bacterial community, and metabolism in goats, leading to a greater abundance of *Prevotella*_1 and *Butyrivibrio*_2 and higher concentrations of beneficial metabolites such as butyric acid; at the same time, there was a decrease of metabolites that hinder nutrient metabolism and represent health hazards. In summary, the present study highlighted that the cassava foliage diet had a positive effect on the goats’ rumen micro-environment while also, in turn, improving antioxidant and immunity capacity and then promoting growth performance. However, the molecular mechanism of cassava foliage promoting the production of beneficial microorganisms and metabolites in the rumen and their impact on intestinal barrier function were worth exploring.

## Figures and Tables

**Figure 1 microorganisms-11-02320-f001:**
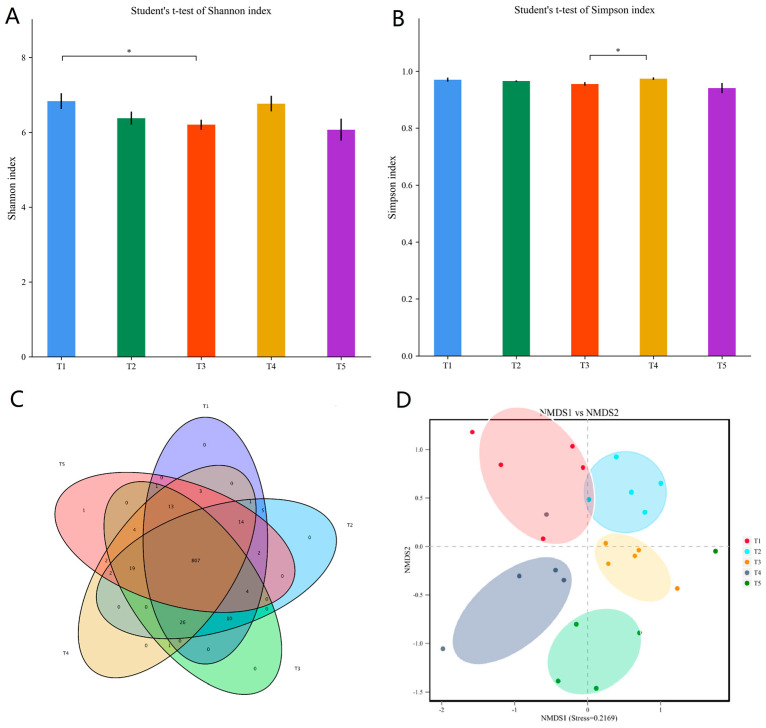
Diversity and community structure of the rumen microbiome in goats fed diets varying in the proportion of cassava foliage. T1, 100% king grass; T2, 25% cassava foliage silage with 75% king grass; T3, 50% cassava foliage silage with 50% king grass; T4, 75% cassava foliage silage with 25% king grass; T5, 100% cassava foliage. (**A**) Shannon diversity index. (**B**) Simpson diversity index. (**C**) Venn diagram of OTUs. (**D**) Non-metric multidimensional scaling (NMDS) analysis of the rumen microbiome. * Means significantly different (*p* < 0.05).

**Figure 2 microorganisms-11-02320-f002:**
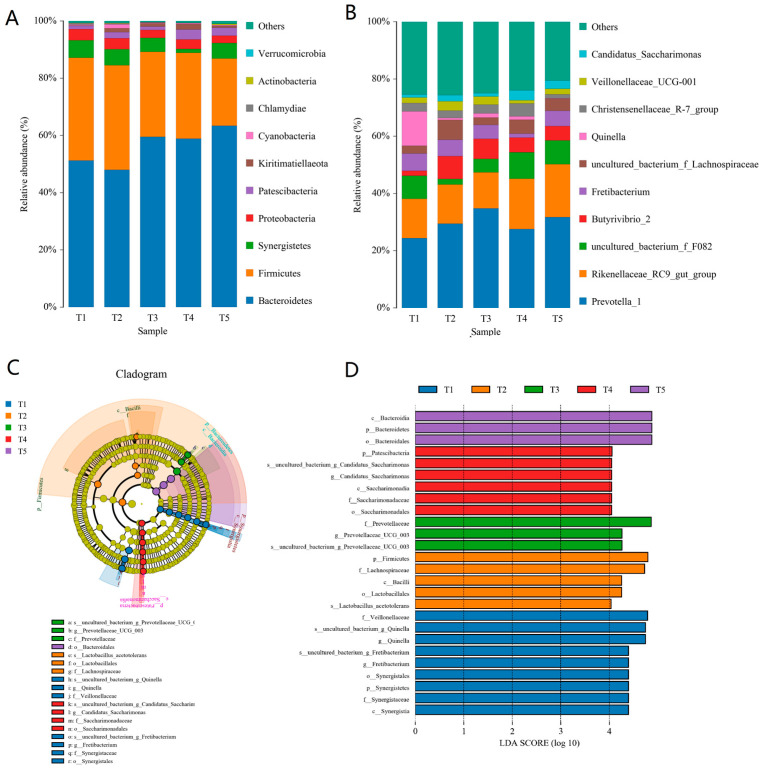
Microbial community composition (phylum, (**A**) and genus, (**B**)) and comparison of microbial communities (via the LEfSe tool) of rumen microbiomes in goats fed diets with varying proportions of cassava foliage (**C**,**D**). T1, 100% king grass; T2, 25% cassava foliage silage with 75% king grass; T3, 50% cassava foliage silage with 50% king grass; T4, 75% cassava foliage silage with 25% king grass; T5, 100% cassava foliage.

**Figure 3 microorganisms-11-02320-f003:**
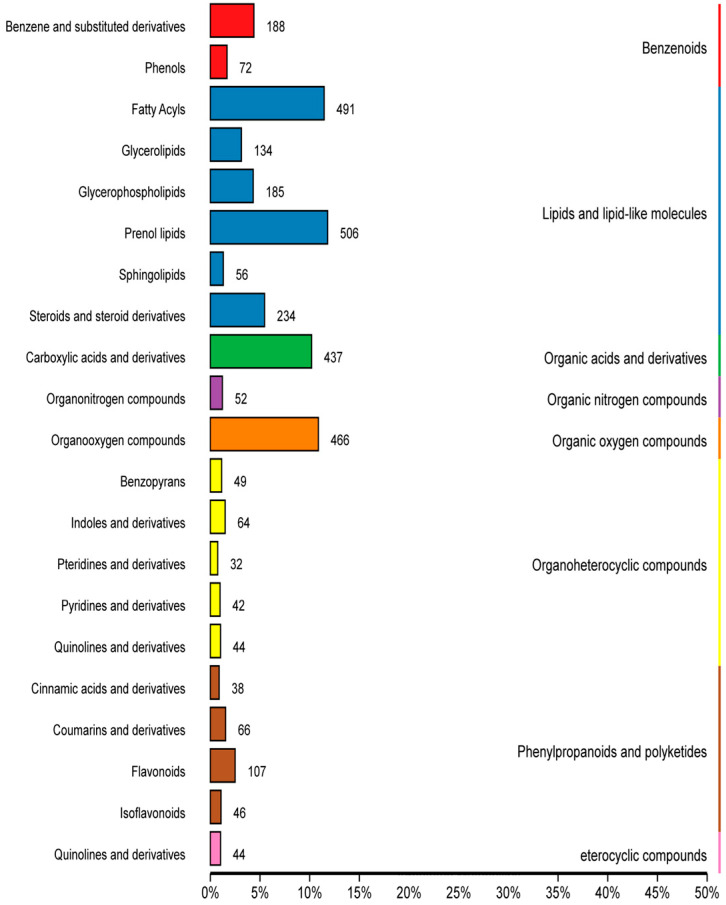
Most common (top 20) metabolites identified in the rumens of goats fed diets, including cassava foliage. Column length represents the number of metabolites in this classification.

**Figure 4 microorganisms-11-02320-f004:**
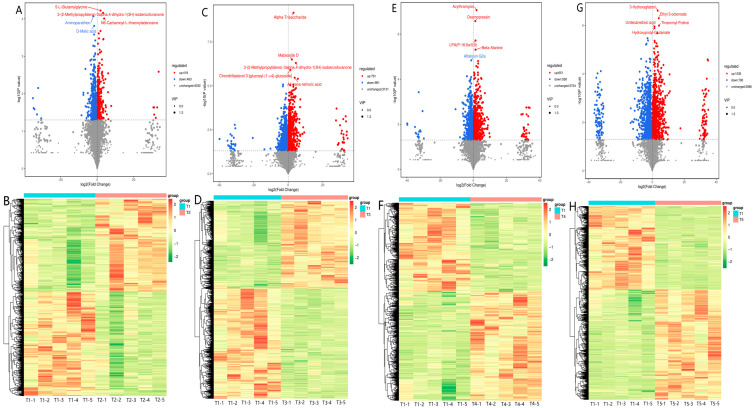
Comparative metabolomics analysis of goat rumens in animals fed diets with varying proportions of cassava foliage. T1, 100% king grass; T2, 25% cassava foliage silage with 75% king grass; T3, 50% cassava foliage silage with 50% king grass; T4, 75% cassava foliage silage with 25% king grass; T5, 100% cassava foliage. Volcano plots show the number of differentially expressed metabolites among treatments ((**A**), T1 vs. T2; (**C**), T1 vs. T3; (**E**), T1 vs. T4; (**G**), T1 vs. T5). Heat maps show the differences in metabolic profiles for goat rumens from different groups ((**B**), T1 vs. T2; (**D**), T1 vs. T3; (**F**), T1 vs. T4; (**H**), T1 vs. T5).

**Figure 5 microorganisms-11-02320-f005:**
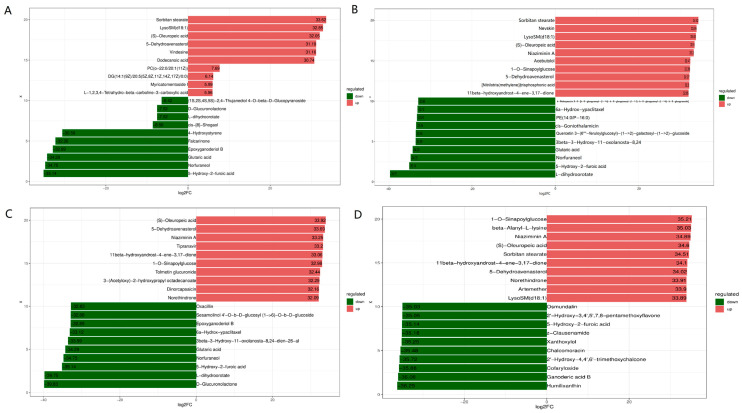
Fold changes in differentially expressed metabolites in goat rumens from animals fed diets with different amounts of cassava foliage ((**A**), T1 vs. T2; (**B**), T1 vs. T3; (**C**), T1 vs. T4; (**D**), T1 vs. T5). T1, 100% king grass; T2, 25% cassava foliage silage with 75% king grass; T3, 50% cassava foliage silage with 50% king grass; T4, 75% cassava foliage silage with 25% king grass; T5, 100% cassava foliage.

**Figure 6 microorganisms-11-02320-f006:**
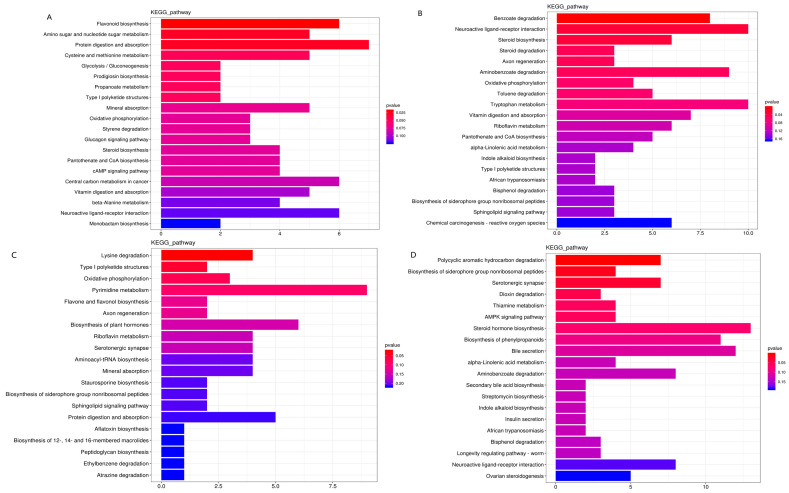
KEEG pathway enrichment analysis of the differentially expressed metabolites observed in goat rumens from animals fed diets with different amounts of cassava foliage ((**A**), T1 vs. T2; (**B**), T1 vs. T3; (**C**), T1 vs. T4; (**D**), T1 vs. T5). T1, 100% king grass; T2, 25% cassava foliage silage with 75% king grass; T3, 50% cassava foliage silage with 50% king grass; T4, 75% cassava foliage silage with 25% king grass; T5, 100% cassava foliage.

**Table 1 microorganisms-11-02320-t001:** Ingredients and nutrient composition of the concentrated goat feed, cassava foliage, and king grass forage (all units g/kg, as feed).

Ingredients	T1	T2	T3	T4	T5	Cassava Foliage	King Grass
Maize	335	335	335	335	335		
Soybean meal	90	90	90	90	90		
Wheat bran	49	49	49	49	49		
Salt	7	7	7	7	7		
Shell powder	7	7	7	7	7		
Sodium bicarbonate	7	7	7	7	7		
Premix ^a^	5	5	5	5	5		
Cassava foliage	0	125	250	375	500		
King grass	500	375	250	125	0		
Nutrition							
ME (MJ/kg) ^b^	9.2	9.2	9.3	9.2	9.3		
Crude protein (g/kg) ^b^	166	168	168	172	176	172	114
Neutral detergent fiber (g/kg) ^b^	448	440	433	428	422	289	658
Acid detergent fiber (g/kg) ^b^	281	275	270	267	260	250	333
Calcium (g/kg) ^b^	8.0	7.9	8.3	8.2	8.0	8.9	1.7
Phosphorus (g/kg) ^b^	2.5	2.2	2.2	2.3	2.4	3.3	1.0

Note: T1, 100% king grass; T2, 25% cassava foliage silage with 75% king grass; T3, 50% cassava foliage silage with 50% king grass; T4, 75% cassava foliage silage with 25% king grass; T5, 100% cassava foliage. ^a^ The premix provided the following per kilogram of diet: VA 15,000 IU; VD 5000 IU; VE 50 mg; Fe 9 mg; Cu 12.5 mg; Zn 100 mg; Mn 130 mg; Se 0.3 mg; I 1.5 mg; and Co 0.5 mg. ^b^ Analyzed values.

**Table 2 microorganisms-11-02320-t002:** Effects of different diets on the growth performance of Hainan black goats.

Treatment	T1	T2	T3	T4	T5	SEM	*p*-Value	
							T	L
Average daily gain (g/day)	32.2 ± 4.9 ^b^	52.3 ± 4.5 ^a^	53.5 ± 1.2 ^a^	51.8 ± 4.0 ^a^	59.5 ± 3.9 ^a^	4.6	*p* < 0.05	*p* > 0.05
Average daily feed intake (g/day)	503 ± 9.6 ^b^	504 ± 11.7 ^b^	532 ± 10.6 ^a^	485 ± 20.8 ^b^	499 ± 21.8 ^b^	7.6	*p* < 0.05	*p* > 0.05
Feed conversion rate	16.9 ± 2.2 ^a^	9.9 ± 0.86 ^b^	10.0 ± 0.37 ^b^	9.5 ± 0.71 ^b^	8.5 ± 0.58 ^b^	1.5	*p* < 0.05	*p* > 0.05

Note: T1, 100% king grass; T2, 25% cassava foliage silage with 75% king grass; T3, 50% cassava foliage silage with 50% king grass; T4, 75% cassava foliage silage with 25% king grass; T5, 100% cassava foliage. T, treatment; L, linear; SEM, standard error of the mean. Means within the same column with different letters are significantly different (*p* < 0.05).

**Table 3 microorganisms-11-02320-t003:** Effects of different diets on blood biochemical indexes in Hainan black goats.

Blood Element	T1	T2	T3	T4	T5	SEM	*p*-Value	
							T	L
TP (g/L)	69.1 ± 2.7	69.2 ± 1.5	75.2 ± 4.3	73.0 ± 1.8	73.2 ± 2.9	1.2	*p* > 0.05	*p* > 0.05
ALB (g/L)	31.7 ± 1.1	30.9 ± 0.86	31.4 ± 0.72	31.7 ± 0.99	26.8 ± 5.1	0.9	*p* > 0.05	*p* > 0.05
BU (mmol/L)	4.8 ± 0.39	5.0 ± 0.31	3.6 ± 0.72	4.2 ± 0.86	4.4 ± 0.96	0.2	*p* > 0.05	*p* > 0.05
CRE (g/L)	59.4 ± 6.5 ^a^	55.8 ± 2.2 ^a^	39.8 ± 7.0 ^b^	42.3 ± 7.7 ^b^	37.1 ± 8.2 ^b^	4.5	*p* < 0.05	*p* > 0.05
UA (μmol/L)	114 ± 30.0	159 ± 34.8	114 ± 73.0	171 ± 65.0	178 ± 60.3	13.9	*p* > 0.05	*p* > 0.05
GL (mmol/L)	2.4 ± 0.36	2.4 ± 0.22	1.7 ± 0.73	1.7 ± 0.82	1.9 ± 0.79	0.2	*p* > 0.05	*p* > 0.05
TCHO (mmol/L)	2.2 ± 0.25	2.0 ± 0.25	1.9 ± 0.48	2.0 ± 0.5	1.7 ± 0.44	0.1	*p* > 0.05	*p* > 0.05
TG (mmol/L)	2.9 ± 0.86 ^a^	0.57 ± 0.04 ^b^	0.55 ± 0.06 ^b^	0.55 ± 0.08 ^b^	0.53 ± 0.08 ^b^	0.5	*p* < 0.05	*p* > 0.05
ALT (IU/L)	37.7 ± 2.8 ^a^	33.8 ± 1.4 ^ab^	33.1 ± 1.7 ^ab^	31.9 ± 3.8 ^ab^	25.6 ± 4.9 ^b^	2.0	*p* < 0.05	*p* > 0.05
AST (IU/L)	97.0 ± 3.7 ^a^	92.1 ± 6.5 ^a^	89.3 ± 3.6 ^a^	76.2 ± 8.1 ^ab^	56.7 ± 13.1 ^b^	7.2	*p* < 0.05	*p* > 0.05

Note: T1, 100% king grass; T2, 25% cassava foliage silage with 75% king grass; T3, 50% cassava foliage silage with 50% king grass; T4, 75% cassava foliage silage with 25% king grass; T5, 100% cassava foliage. T, treatment; L, linear; SEM, standard error of the mean. Means within the same column with different letters are significantly different (*p* < 0.05).

**Table 4 microorganisms-11-02320-t004:** Dietary effects on four antioxidant indexes in Hainan black goats.

Treatment	T1	T2	T3	T4	T5	SEM	*p*-Value	
							T	L
T-AOC (U/mL)	21.7 ± 0.96 ^bc^	24.2 ± 1.0 ^ab^	24.8 ± 0.50 ^a^	20.0 ± 0.95 ^c^	15.9 ± 0.79 ^d^	1.6	*p* < 0.05	*p* > 0.05
MDA (nmol/mL)	11.8 ± 0.59 ^ab^	10.4 ± 0.62 ^b^	8.4 ± 0.54 ^c^	12.7 ± 0.36 ^a^	13.0 ± 0.49 ^a^	0.8	*p* < 0.05	*p* > 0.05
T-SOD (U/mL)	1881 ± 56.0 ^c^	2211 ± 54.2 ^b^	2428 ± 53.5 ^a^	1830 ± 56.6 ^c^	1555 ± 113.4 ^d^	152	*p* < 0.05	*p* > 0.05
GSH-P (U/mL)	397 ± 8.7 ^b^	434 ± 18.2 ^b^	481 ± 8.7 ^a^	348 ± 9.1 ^c^	321 ± 15.3 ^c^	28.7	*p* < 0.05	*p* > 0.05

Note: T1, 100% king grass; T2, 25% cassava foliage silage with 75% king grass; T3, 50% cassava foliage silage with 50% king grass; T4, 75% cassava foliage silage with 25% king grass; T5, 100% cassava foliage. T, treatment; L, linear; SEM, standard error of the mean. Means within the same column with different letters are significantly different (*p* < 0.05).

**Table 5 microorganisms-11-02320-t005:** Effects of different diets on immune indexes in Hainan black goats.

Treatment	T1	T2	T3	T4	T5	SEM	*p*-Value	
							T	L
IgA (mg/L)	2842 ± 75.7 ^bc^	3057 ± 67.0 ^ab^	3202 ± 94.9 ^a^	2828 ± 73.3 ^c^	2267 ± 48.2 ^d^	158	*p* < 0.05	*p* > 0.05
IgG (g/L)	19.0 ± 0.51 ^b^	20.1 ± 0.71 ^b^	22.2 ± 0.64 ^a^	16.8 ± 0.32 ^c^	15.4 ± 0.83 ^c^	1.2	*p* < 0.05	*p* > 0.05
IgM (mg/L)	2120 ± 88.3 ^b^	2539 ± 78.3 ^a^	2585 ± 68.2 ^a^	1778 ± 56.7 ^c^	1540 ± 57.5 ^d^	431	*p* < 0.05	*p* > 0.05
C3 (mg/L)	665 ± 32.9 ^a^	681 ± 29 ^a^	714 ± 13.1 ^a^	563 ± 27.4 ^b^	543 ± 24.1 ^b^	33.8	*p* < 0.05	*p* > 0.05
C4 (mg/L)	317 ± 25.9 ^bc^	341 ± 21.4 ^b^	424 ± 19.0 ^a^	295 ± 15.9 ^bc^	264 ± 12.2 ^c^	27.1	*p* < 0.05	*p* > 0.05

Note: T1, 100% king grass; T2, 25% cassava foliage silage with 75% king grass; T3, 50% cassava foliage silage with 50% king grass; T4, 75% cassava foliage silage with 25% king grass; T5, 100% cassava foliage. T, treatment; L, linear; SEM, standard error of the mean. Means within the same column with different letters are significantly different (*p* < 0.05).

**Table 6 microorganisms-11-02320-t006:** Effects of different diets on rumen pH and volatile fatty acids in Hainan black goats (all units mol/100 mol).

Treatment	T1	T2	T3	T4	T5	SEM	*p*-Value	
							T	L
pH	7.1 ± 0.06	7.2 ± 0.08	7.1 ± 0.10	7.2 ± 0.12	7.2 ± 0.14	0.02	*p* > 0.05	*p* > 0.05
Acetate	54.6 ± 4.7	56.2 ± 1.8	56.4 ± 9.4	57.0 ± 4.0	54.6 ± 4.2	0.49	*p* > 0.05	*p* > 0.05
Propionate	10.9 ± 0.54	10.4 ± 0.65	9.2 ± 2.0	10.8 ± 0.37	10.6 ± 0.48	0.31	*p* > 0.05	*p* > 0.05
Isobutyrate	1.0 ± 0.15	0.90 ± 0.05	0.92 ± 0.24	0.90 ± 0.12	1.1 ± 0.12	0.04	*p* > 0.05	*p* > 0.05
Butyrate	5.5 ± 0.95 ^b^	6.6 ± 1.2 ^a^	6.0 ± 1.9 ^ab^	5.7 ± 0.9 ^b^	3.9 ± 0.79 ^c^	0.45	*p* < 0.05	*p* > 0.05
Isovalerate	27.6 ± 4.8	25.5 ± 1.9	27.0 ± 8.3	25.3 ± 4.4	29.6 ± 4.7	0.78	*p* > 0.05	*p* > 0.05
Valerate	0.45 ± 0.09	0.41 ± 0.01	0.44 ± 0.19	0.38 ± 0.06	0.38 ± 0.03	0.01	*p* > 0.05	*p* > 0.05

Note: T1, 100% king grass; T2, 25% cassava foliage silage with 75% king grass; T3, 50% cassava foliage silage with 50% king grass; T4, 75% cassava foliage silage with 25% king grass; T5, 100% cassava foliage. T, treatment; L, linear; SEM, standard error of the mean. Means within the same column with different letters are significantly different (*p* < 0.05).

## Data Availability

The data presented in this study are available upon request from the corresponding author.

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
