# Peer review of "Cassava Foliage Effects on Antioxidant Capacity, Growth, Immunity, and Ruminal Microbial Metabolism in Hainan Black Goats"

_microorganisms, 2023, doi:10.3390/microorganisms11092320_

Round 1

Reviewer 1 Report

This manuscript is generally OK, although it seems that a lot if its results have been previously found with other animals. Still, it is very thorough, and reaches a reasonable conclusion. I marked down quality of presentation for two reasons: a) Figures 2-6 are very hard to read even when magnified b) The manuscript does not properly follow the Rule of the Decimal Place. If there are no numbers to the left of the decimal point, you can present up to two numbers to the right of the decimal One or two numbers to the left of the decimal allows one digit to the right of the decimal. Three or more digits to the left means no digits to the right of the decimal, and in fact no decimal is needed in that case. 

Also, the phrase "in this study", which means nothing, really, appears 8 times. Just present the facts...

Author Response

Thank you for your kindly and professional comment.Your comments are very helpful to improve the quality of our manuscript.

1.Figures 2-6 are very hard to read even when magnified. 
Response: We try our best to improve the clarity of the images and change the layout for easier reading.

  1. The manuscript does not properly follow the Rule of the Decimal Place.

Response: We have modified the data style in the table according to your suggestions.

3.Also, the phrase "in this study", which means nothing, really, appears 8 times. Just present the facts.

Response: We have revised the relevant descriptions in the manuscript.

Reviewer 2 Report

Comments to editor

microorganisms-2561192

this paper  exhibits the effects of feeding goats with cassava forage on the immunity, antioxidant and microbial diversity. This work is traditional because there are a huge data regarding this topics.

However, there is a lack data regarding its effects on microbiota in goats.  Authors need to make their results more attractive.

I suggest extensive  major revision

What is the novelty of this work?

There are more than 30 papers about the using of Cassava in goat diets?

The abstract doesn’t contain the experimental design?  

The addition of dietary cassava foliage als……….. in which level?

Comments

-        Line 15 not bold

-        The references in the whole manuscript are not corrected.

-        Line20, . Feeding a diet containing cassava foliage resulted…… how about the level?

-        Immunity and antioxidant indices should separated in the abstract.

-        Line 27-28, this sentence is general ?

-        The keywords should be not similar with the words of the title.

-        Line 40, , the rapid development of ruminant husbandry in China has led to a shortage of high-quality forage? How?

-        Lines 46-47, this sentence should rewrite.

-        Line 77, from tropical China. What does it means?

-        Line 88, NRC (2007)??????? Where is the references?

-        How the FCR, body weight, feed intakes were calculated?

-         

-        Lines 102, how blood samples were collected? There was anesthesia? Veins name? how blood samples treated, centrifugation, EDTA tubes or NO, plasma or serum ? Please add more data?

-         

-        Line 164, Statistics ???

-        The statistical model should be revised ?

-        the Levene and Shapiro–Wilk test should applied for deterimine the normality of data?

-        Lines 174 – 176 This is incorrect; please revise this sentence and provide a clear message.

-        Line 180 should add the significant letters such a,b,c  apply in all the tables.

-        In table 2, FCR is very high in the control group?

-        Discussion  should separated

-        Material and methods are lack regarding to the measurements of biochemical indices, kits, company, method….etc.

-        I think authors should focus on the impacts of cassava feeding on rumen microbiota.

-        Conclusion: should reflect the most important results,  suitable level,  recommended, mechanisms. Future studies

-         

-         

-         

-         

moderate English editing  

Author Response

Thank you for your kindly and professional comment.Your comments are very helpful to improve the quality of our manuscript.

General comments

  1. What is the novelty of this work?

Response: We comprehensive reported the effects of cassava foliage on the antioxidant activity, immunity status, rumen microbial and metabolism.

  1. There are more than 30 papers about the using of Cassava in goat diets?

Response:

  1. The abstract doesn’t contain the experimental design?  

Response: We had added these content.

  1. The addition of dietary cassava foliage als……….. in which level?

Response: Five level: 0% (T1), 25% (T2), 50% (T3), 75% (T4) and 100% (T5) of the cassava foliage silage replaced king grass.

Comments:
   Line 15 not bold

Response: We have revised it.

-        The references in the whole manuscript are not corrected.

Response: We have made modifications based on the template.

-        Line20, . Feeding a diet containing cassava foliage resulted…… how about the level?

-      Response:  Feeding a diet containing 50% cassava foliage resulted in beneficial effects

 Immunity and antioxidant indices should separated in the abstract.

-    Response: We have separated the two parts.

    Line 27-28, this sentence is general ?

 Response: We have revised the sentence.

-        The keywords should be not similar with the words of the title.

-      Response: We have revised the keywords.

   Line 40, , the rapid development of ruminant husbandry in China has led to a shortage of high-quality forage? How?

-     Response: We provided a more detailed explanation.

    Lines 46-47, this sentence should rewrite.

-   Response: We have rewrote this sentence.

     Line 77, from tropical China. What does it means?

-      -   Response: We corrected as “tropical regions of China”

  Line 88, NRC (2007)??????? Where is the references?

Response: We added the references as follow.

National Research Council (NRC). (2007). Nutrient requirements of small ruminants: Sheep, goats, cervids, and new world camelids. Washington, DC: National Academies Press.

    How the FCR, body weight, feed intakes were calculated?

-         Response: We added the details in Animal, diet and sample.The measurement of average daily gain (ADG), average daily feed intake (ADFI) and the feed conversion ratio (FCR) were conducted as previously described, the details as follow: each goat was weighed at the first and last day of treatment period to calculate ADG, amounts of feed provide and residual of each goat were recorded daily During the treatment period to calculate DMI, and the FCR was the ratio of ADFI:ADG

-        Lines 102, how blood samples were collected? There was anesthesia? Veins name? how blood samples treated, centrifugation, EDTA tubes or NO, plasma or serum ? Please add more data?

-         Response:We added the details in Animal, diet and sample. About 4 mL blood samples were taken from each goats jugular vein applying vacutainer(No additive) before morning feeding at 60 day of treatment period. Above samples  centrifugation at 3,000×g for 20 min at 4°C, and the serum was separated into three parts, stored at −20°C for biochemical indexes, antioxidant capacity and immunity factors analyses.

    Line 164, Statistics ???

-      Response: We corrected as Statistical analysis.

  The statistical model should be revised ?

- Response: We revised statistical model. The orthogonal polynomial contrasts (linear and quadratic) were used to evaluate the effect of cassava foliage inclusion ratio in goat dietary.

-        the Levene and Shapiro–Wilk test should applied for deterimine the normality of data?

- Response: After the Shapiro-Wilk test, all data in this study distributions accord with normal distribution.

 Lines 174 – 176 This is incorrect; please revise this sentence and provide a clear message.

-    Response: We corrected as “The ADFI was highest in T3 treatments group (P< 0.05), and shown a trend of first increasing and then decreasing”. 

    Line 180 should add the significant letters such a,b,c  apply in all the tables.

-    Response: We have added the significant letters in all the tables.

    In table 2, FCR is very high in the control group?

-      Response: In this study, the FCR was ADFI/ADG, and the higher FCR numerical value mean lower feed conversion efficiency, four cassava foliage treatment had lower FCR, indicated high feed conversion efficiency.

  Discussion  should separated

-    Response: We have separated the Results and Discussion part.

    Material and methods are lack regarding to the measurements of biochemical indices, kits, company, method….etc.

Response: We added the details in Chemical analysis.

       I think authors should focus on the impacts of cassava feeding on rumen microbiota.

-  Response: We have made some modifications to the abstract and conclusion.

      Conclusion: should reflect the most important results,  suitable level,  recommended, mechanisms. Future studies

Response: We have revised the conclusion.